# Belantamab Mafodotin: From Clinical Trials Data to Real-Life Experiences

**DOI:** 10.3390/cancers15112948

**Published:** 2023-05-27

**Authors:** Sonia Morè, Massimo Offidani, Laura Corvatta, Maria Teresa Petrucci, Francesca Fazio

**Affiliations:** 1Clinica di Ematologia Azienda Ospedaliero, Universitaria delle Marche, 60126 Ancona, Italy; sonia.more@live.it; 2Unità Operativa Complessa di Medicina, Ospedale Profili, 60044 Fabriano, Italy; laura.corvatta@sanita.marche.it; 3Hematology, Department of Translational and Precision Medicine, Azienda Policlinico Umberto I, Sapienza University of Rome, 00185 Rome, Italy; petrucci@bce.uniroma1.it (M.T.P.); fazio@bce.uniroma1.it (F.F.)

**Keywords:** multiple myeloma, belantamab mafodotin, antibody–drug conjugate

## Abstract

**Simple Summary:**

This paper deals with Belantamab mafodotin, a novel anti-BCMA antibody-drug conjugate licensed by regulatory agencies for the treatment of anti-CD38-refractory multiple myeloma patients. In addition to describing the mechanism of action and general information about the drug, this review mostly focuses on real-life data from American, Asian and European experiences on it. The aim of this overview is to picture an updated scenario on Belantamab mafodotin in different completed or ongoing clinical trials, studying it in different settings. Furthermore, this paper would like to provide precise data collection about the efficacy and safety of Belantamab mafodotin real-world data, showing the reproducibility of randomized clinical trial data that is also in real-life settings.

**Abstract:**

Despite the recent approval of novel immunotherapies, such as immunomodulatory drugs, proteasome inhibitors and anti-CD38 monoclonal antibodies, Multiple Myeloma (MM) remains incurable, and the acquisition of triple-refractoriness leads to really dismal outcomes in even earlier lines of therapy. More recently, innovative therapeutic strategies targeting B cell maturation antigen (BCMA), highly expressed on the plasma cell surface, are drawing different future landscapes in terms of effectiveness and outcomes. Belantamab Mafodotin, a first-in-class anti-BCMA antibody–drug conjugate, demonstrated good efficacy and safety profile in triple-refractory patients in the phase 2 DREAMM-2 trial, and it was approved for the treatment of MM triple-exposed patients with >4 prior lines of therapy. Here, starting from Belantamab Mafodotin clinical trials and also exploring combination studies and different schedules in order to improve its efficacy and toxicity, we focused on real-life experiences all over the world, which have confirmed clinical trial data and encourage further Belantamab Mafodotin investigations.

## 1. Introduction

Multiple myeloma (MM) is the second most common hematological disease accounting for approximately 10% of all hematological malignancies, and its typically an elderly hematologic disease, with the median age of patients at the diagnosis of about 65 years [1].

MM is characterized by abnormal plasma cell proliferation in the bone marrow, which produces an excess of monoclonal protein (M-protein) detected in blood and urine. This M-protein causes specific organ damage resulting in MM signs and symptoms, typically hypercalcemia, renal insufficiency, anemia and osteolytic bone lesions [2]. 

The therapeutical landscape of effective anti-myeloma drugs has widely expanded in the last decade thanks to the introduction of proteasome inhibitors (PIs, such as bortezomib, carfilzomib and ixazomib), immunomodulatory drugs (IMiDs, such as thalidomide, lenalidomide and pomalidomide) and, recently, monoclonal antibodies (mAbs), such as elotuzumab, daratumumab and isatuximab [2]. The introduction of these agents has translated into prolonged progression-free survival (PFS) as well as overall survival (OS) with significantly fewer toxicities and improved quality of life. Currently, thanks to the therapeutic first-line approaches, including quadruplet induction combination, high-dose chemotherapy, followed by autologous hematopoietic stem-cells transplantation (ASCT), and consolidation–maintenance, the 10-year OS probability is about 60% in patients eligible for ASCT. [3]. In non-transplant eligible (NTE) patients, therapeutical first-line approaches based on doublet or triplet combination with PIs, IMiDs and mAbs result in a prolonged OS, with a median OS of 5 years. 

However, MM still remains largely an incurable disease with poor outcomes, especially among patients who become resistant to therapies [4]. 

Immunotherapeutic approaches, both passive, for example, mAbs and cellular products targeting clonal plasma cells, or active (when a patient’s immune system is stimulated to induce an immune response against plasma cells), have been investigated to harness the patients’ immune system to destroy clonal plasma cells and, nowadays, represent an effective strategy for the treatment of MM [2]. 

Clonal plasma cells express several antigens on their surface, with the most important molecule under investigation as potential targets for mAbs, such as CD38, CD40, CD138, CD56, CD54, PD1, kappa light chain, B-cell maturation antigen (BCMA) and the signaling lymphocyte activation molecule F7 (SLAMF7). To date, novel immunotherapies, including antibody–drug conjugates (ADC), bispecific antibodies and chimeric antigen receptor T-cell (CAR-T), represent an important option for relapsed or refractory disease resulting in high-quality, deep and durable responses [5].

Several mechanisms of action are described for mAbs therapies: direct cytotoxic effects on the clonal plasma cell, antibody-dependent cellular cytotoxicity (ADCC), antibody-dependent cell-mediated phagocytosis (ADCP) and complement-dependent cytotoxicity (CDC) [6].

Recently, these immunotherapy approaches have been investigated in monotherapy, firstly, and then in combination with other effective anti-plasma cell agents to improve outcomes. B-cell maturation antigen (BCMA), a member of the tumor necrosis factor (TNF) superfamily, represents a promising new target for MM therapy [7]. As more in detail described below, belantamab mafodotin is the first-in-class ADC targeting BCMA approved for RR/MM patients. Specifically, SLAMF7 and CD38 are two antibody targets of particular interest in MM. Elotuzumab, a humanized immunoglobulin IgG_1_ monoclonal antibody binding SLAMF7, expressed both on MM and NK cells, is the first in class mAb to be explored in MM and to show, in combination with lenalidomide and dexamethasone (Rd), a significant OS benefit in RRMM compared to Rd in phase II ELOQUENT-2 trial [8]. Furthermore, elotuzumab has been investigated in association with pomalidomide and dexamethasone (Pd), resulting in superior outcomes compared with standard therapy Pd [9].

Daratumumab, the first-in-class humanized IgG_1_K monoclonal antibody targeting CD38, exerts its specific mechanisms of action through different patterns: CDC, ADCC, ADCP, induction of direct plasma cell apoptosis, and through the modulation of CD38 enzyme activities [10]. After initial approval in heavily pretreated patients as monotherapy [11], daratumumab has been approved in combination with Rd (DRd) and Vd (DVd) for the treatment of RRMM patients who have received at least one prior line of therapy [12,13]. Recently, daratumumab has been investigated in RRMM patients in combination with Pd or in combination with carfilzomib and dexamethasone (DKd) in phase III APOLLO and CANDOR trials [14,15]. 

More recently, based on the results of the pivotal phase III MAIA trial, daratumumab was approved, in combination with Rd, in newly diagnosed MM not eligible for ASCT [16]. Another daratumumab-based combination therapy has been recently approved, D-VMP, for newly diagnosed MM patients not eligible for ASCT [17]. Based on the results of the phase III CASSIOPEIA study, the new standard of care for newly diagnosed transplant-eligible MM patients is daratumumab in combination with VTd (bortezomib, thalidomide and dexamethasone) [18]. 

Isatuximab, an IgG-k chimeric monoclonal antibody targeting a specific epitope on CD38, was investigated in combination with other drugs in relapsed/refractory settings. Based on the efficacy results, in terms of ORR and PFS, of the phase III ICARIA study, the combination of isatuximab plus Pd was approved for RRMM patients who have received at least two prior lines of therapy, including lenalidomide [19]. Recently, the phase III pivotal trial IKEMA demonstrated superior outcomes in terms of PFS for patients treated with isatuximab in combination with carfilzomib and dexamethasone (Kd) compared to patients treated with Kd [20]. Compared to daratumumab, isatuximab presents a different mechanism: firstly, it has been demonstrated that it is able to induce direct apoptosis in MM cells. Next, it is able to inhibit the enzymatic activity of CD38 more than daratumumab and is less effective in exerting CDC. [21]. Unfortunately, as with other agents, resistance to anti-CD38 monoclonal antibody therapy happened. To date, no standard of care has been established for RRMM patients who have been exposed to the three main classes of anti-myeloma drugs, and these patients have limited treatment options representing an unmet medical need. The outcome of patients failing standard care regimens, defined as triple refractory (including PIs, IMiDs and mAbs), is poor, with a median PFS of 5 months and a median OS of 12 months [22]. For this purpose, there is a very urgent need to develop and introduce novel therapeutical approaches.

Selinexor, melfulflen, cerebron E3 ligase modulators (CELMoDs) and venetoclax represent very promising drugs for RRMM patients’ treatment.

The first-in-class oral selective inhibitor of nuclear export targeting exportin-1, selinexor, received FDA approval in combination with bortezomib and dexamethasone (SVd) for RRMM patients after the first line of therapy [23]. The median PFS for patients treated with SVd was 13.9 months compared to 9.5 months for patients treated with Vd. In addition, selinexor was approved in combination with dexamethasone for triple refractory patients [24]. Melfuflen (melfuflen-flufenamide) is the first-in-class peptide-drug conjugate targeting aminopeptidases and releasing alkylating agents into clonal plasma cells [25]. The phase II HORIZON trial demonstrated melfuflen efficacy in association with dexamethasone, with a median PFS of 4.2 months in a very heavily pretreated population [26]. Based on these results, it has been recently approved by FDA, in combination with dexamethasone, in patients who received at least four previous lines of therapy. Iberdomide, a first-in-class CELMoDs, was an advanced IMiDs showing more efficacy compared to lenalidomide and pomalidomide [27]. Lastly, venetoclax, the oral BCL-2 targeted therapy used in CLL and AML, showed promising results in RRMM patients harboring t(11;14) or with high BCL-2 expression [28].

Frequent, deep, and durable responses to BCMA-directed CAR-T cells (idecabtagene–vicleucel) have been recently reported in MM patients who were triple-class exposed and refractory to their last regimen in the multicenter pivotal phase II KarMMa-2 trial [29]. This has led to the rapid approval by FDA and by EMA of the first product (idecabtagene–vicleucel) for RR MM patients after three lines of therapy. On February 2022, FDA approved a second product, ciltacabtagene–autoleucel, for the treatment of RR MM patients after four or more lines of therapy, based on the results of the phase 1b/2 study CARTITUDE-1 [30]. Results from subcutaneous teclistamab, a T-cell redirecting bispecific antibody targeting BCMA, showed an impressive overall response rate (ORR) of 73% in heavily pretreated patients with a median of five prior lines of therapy [31].

## 2. Belantamab Mafodotin

Belantamab Mafodotin is the first-in-class ADC targeting BCMA that obtained FDA accelerated approval in August 2020, based on the results of phase II DREAMM-2 trial, for patients with RRMM who have received at least four prior lines of therapy, including a proteasome inhibitor, an immunomodulatory agent and an anti-CD38 monoclonal antibody [32]. On November 2022, FDA announced that the process for the withdrawal of the US marketing authorization for belantamab mafodotin had been initiated. This action is based on the results of the DREAMM-3 trial (NCT04162210), in which the primary endpoint of PFS was not met, with a hazard ratio of 1.03 (95% confidence interval, 0.72–1.47) in a head-to-head comparison of belantamab mafodotin versus Pd. A longer observed median PFS of 11.2 months was seen with belantamab mafodotin, compared with 7 months for Pd. 

### 2.1. Mechanism of Action

ADCs, an evolution of naked mAbs, consist of a mAb attached to a specific cytotoxic payload covalently conjugated through chemical linkers. The mechanism of action of ADCs is unique, whereby the mAb binds a tumor-specific antigen on clonal plasma cells resulting in cytotoxic payload internalization. Once the ADCs are intracellular, lysosomal degradation occurs, causing the release of the toxic payload within the plasma cells. Then, the free toxic payload enters the cytoplasm and/or nucleus, exerting its effect and causing apoptosis and cell death [33]. The first ADC approved by FDA for use in hematological malignancies was brentuximab vedotin in 2018. This ADC targets CD30, and it has been approved for Hodgkin’s lymphoma selecting CD30 expressing T-cell lymphomas [34].

BCMA is a member of the TNF receptor superfamily and can be an optimal target for ADCs in MM due to its high expression on clonal plasma cells. Indeed, it is a cell-surface receptor protein expressed almost in end-stage B lymphocytes and clonal plasma cells. Because of its unique expression on MM cells, BCMA has become of interest to developing specifically targeted immunotherapies for MM. During the malignant transformation of immature plasma cells, the BCMA receptor and its ligand, named a proliferation-inducing ligand (APRIL) and B-cell activating factor (BAFF), overexpression activate several signal transduction pathways involved in oncogenesis, including nuclear factor kappa-light-chain-enhancer of activated B cells (NF-kB), protein kinase B (AKT), signal transducer and activator of transcription 3 (STAT3), phosphoinositide 3-kinase (PI3K) and mitogen-activated protein kinases (MAPK) cascades. Thus, BCMA overexpression promotes tumor growth, survival and drug resistance within malignant plasma cells. In addition, the **γ**-secretase activity of the membrane receptor is able to release soluble BCMA (sBCMA), which correlates with the phase of MM’s history, increasing through the progression from asymptomatic MGUS stage to smoldering myeloma and then to active MM, and should be helpful to monitor hematological response during treatment phase [7].

The most common naked mAbs used in MM treatment are humanized antibodies or fully human immunoglobulin G subtype thanks to its long-circulating half-life in the bloodstream and lower immunogenicity. For ADCs, the choice of the ideal antibody is important. 

However, for the choice of ideal ADCs, it is important, beyond the antibody structure, the covalent linker because, as mentioned, linkers play a crucial role in releasing the potent drug at target tumor cells. Indeed, it has to be able to avoid premature degradation in the plasma because it can cause the release of cytotoxic payload with off-target effects on healthy cells. But, at the same time, it has to be able to degrade the cytotoxic component in the pathologic cells once the ADC can be internalized in the malignant target cell [33]. ADCs linkers are classified according to different categories in terms of the mechanism of drug release and their stability in circulation, including cleavable linkers and non-cleavable linkers. Cleavable linkers are designed to respond to a specific physiological environment, such as there being high glutathione concentrations, low pH, and special protease, which could assist the linkers in enabling chemical or biochemical reactions by way of hydrolyzation or proteolysis. The second group is non-cleavable linkers that rely on the monoclonal antibody degradation after ADCs’ internalization within the lysosomes and endosomes to generate the metabolites containing the active cytotoxic drugs with or without a portion of the linkers [35]. The choice of the optimal killing molecule is important, too; indeed, it should be able to exert direct cell toxicity, and it should have an adequate site from where the conjugate releases the drug in the specific tumor cell. The most used payload in hematological malignancy is microtubule inhibitor agents such as maytansinoids and auristatin (including monomethyl auristatin E and F), which are able to bind the tubulin, causing G2/M arrest and apoptosis. In addition, DNA-modifying drugs such as calicheamicin, inducing cell death by DNA double-strand breaking, are another common form of payload used [36]. 

Belantamab mafodotin, also called belamaf (GSK2587916), is the first-in-class humanized IgG1 ADC that targets BCMA, approved for RRMM patients after four prior lines of therapy. This ADC is composed of an antibody that is able to bind BCMA, conjugated to a cytotoxic agent, monometil auristatin F (mafodotin), by a protease-resistant maleimidocaproyl linker. After binding BCMA on target cells, Belamaf is internalized and undergoes a process of proteolytic cleavage, releasing mafodotin. The released mafodotin disrupts the microtubular cell network leading to cell cycle arrest and apoptosis [32] (Figure 1). 

Belamaf improves additional anticancer effects such as antibody-dependent cellular toxicity (ADCC) and antibody-dependent cellular phagocytosis via recruitment of the immune system, thanks to afucosylation of the antibody portion. In vitro preclinical studies demonstrated that ADCC is enhanced in combination with IMiDs. In addition, in vitro and in vivo studies showed that Belantamab mafodotin efficacy is characterized by higher T and NK lymphocyte concentration and an increase of markers of immunomediated tumor cell death [37]. 

### 2.2. Dosing and Administration

The recommended dose for belantamab mafodotin is 2.5 mg/kg, administered once every 3 weeks and continued until disease progression or unacceptable toxicity. No dose modification is required for renal impairment if the estimated glomerular filtration rate (eGFR) is >30 mL/min/1.73 m^2^ or for mild hepatic impairment where total bilirubin is ≤1.5 times the upper limit of normal. However, no dosing recommendations have been established for patients with eGFR < 30 mL/min/1.73 m^2^, those with end-stage renal disease either with or without dialysis, or those with moderate to severe hepatic impairment. Therefore, due to the lack of safety and efficacy data for severe renal and hepatic impairment at this time, belamaf should be considered contraindicated in these patients. Dose interruption, reduction or discontinuation may be required for adverse events such as thrombocytopenia, infusion reactions or ocular adverse events. The dose reduction for the first adverse event is belamaf 1.9 mg/kg once every 3 weeks. Belamaf is administered as an intravenous infusion for over 30 min and does not require any premedication. However, if patients experience an infusion reaction, then premedication should be administered for all subsequent infusions [38].

### 2.3. Toxicities

The most important adverse event related to belamaf therapy is ocular toxicity as reported by DREAMM-1 and DREAMM-2 clinical trials. Ocular events include keratopathy (defined as corneal epithelial changes named microcyst-like epithelial changes, MECs), best-corrected visual activity (BCVA) reduction and any other ocular symptoms, such as blurred vision, dry eye and corneal ulceration [32].

ADC adverse events could be explained by on-target or off-target mechanisms: indeed, considering that the majority of the proteins targeted by these agents are not expressed in the cornea (such as BCMA), ocular events may represent a specific off-target mechanism.

Farooq et al. [37] reported that ADC could reach corneal epithelial cells through the vascularized part of the limbus or through the tear film because it has been previously detected in animal tears, circumstantially pointing toward its off-target damage site as the cornea lacks BCMA.

Microtubule-disrupting monomethylauristatin-F (MMAF) is the cytotoxic component of Belamaf that is linked to a monoclonal antibody via protease-resistant maleimidocaproyl (mc) linker. MMAF is proposed as an attributable cause of ocular toxicity along with other ADCs that use the MMAF. Several factors are involved in promoting off-target ocular toxicity by ADCs, such as linker instability or premature cleavage in extracellular environments, linker-cytotoxin intracellular metabolism, and Fc-receptor-mediated cellular uptakes. Belamaf induces apoptosis of myeloma cells but may cause concomitant off-target apoptosis of corneal epithelial cells due to microtubulin inhibition caused by MMAF. MECs require medium to high magnification power when examined via **a** slit lamp. On in-vivo confocal microscopy, these changes may appear as hyper-reflective opacities. When belamaf encroaches the cornea via limbus vasculature or tears, it undergoes the process of solute internalization or cellular uptake via macropinocytosis into the limbal epithelial stem cells. Once belamaf is internalized into corneal cells, it inhibits their proliferation, inducing apoptosis. The second step is the migration of the alive epithelial corneal cells toward the peripheral cornea: at this moment, we can detect MECs using the slit lamp images in the absence of specific ocular symptoms, too. Belamaf-containing cells and MECs are initially found at the periphery of the cornea but eventually migrate in a centripetal fashion and then vanish due to extrusion [39]. Then, when the migration of belamaf-carrying cells reaches the central cornea, ocular symptoms are described by patients [40]. However, the exact mechanism of damage is not well defined and requires further elucidation. Investigation of the mechanism and pharmacokinetics of ocular toxicity is underway, as well as the evaluation of various mitigation and management strategies to prevent and treat this toxicity.

In part-1 of the DREAMM-1 trial, where belamaf dose ranged from 0.03 mg/kg to 4.60 mg/kg, the ocular toxicity occurred more frequently at a larger dose than a smaller dose [41].

In the pivotal phase II DREAMM-2 study, keratopathy was the most common ocular toxicity (73%) irrespective of its grades (71% in 2.5 mg/kg versus 75% in 3.4 mg/kg) and the most common complaints were also blurred vision (22% for 2.5 mg/kg versus 30% for 3.4 mg/kg) and dry eyes (14% for 2.5 mg/kg versus 23% for 3.4 mg/kg). When corneal changes occur in the form of keratopathy, most patients are symptomatic. Nevertheless, the absence of corneal symptoms does not rule out the existence of keratopathy, as detected by the slit lamp and visual acuity testing data of DREAMM-2.

In the dosing cohort of 2.5 mg/kg of DREAMM-2, 72% of patients had MECs and 54% had vision changes. Contrarily, only 25% of those patients reported blurred vision and 15% reported dry eyes. Similarly, in the 3.4 mg/kg cohort of belamaf, 77% of patients had MEC, but 33% of those reported blurred vision and 25% reported dry eyes. These data suggest that ocular toxicity requires active surveillance irrespective of symptoms. Even patients with grade G3 or 4 keratopathies, i.e., severe superficial keratopathy and corneal ulcers, may be asymptomatic. Such patients may continue to receive belamaf with ongoing toxicity unless screened via **a** slit lamp. The median time to onset of MECs was 36 days, and it is longer for other specific ocular symptoms, such as BCVA decline (median time of 64 days), blurred vision (median time of 52 days), and 42 for dry eyes. Globally, 3% of patients experienced ocular events that led to treatment discontinuation. According to DREAMM-2 data, 48% of patients resolved MECs, and 82% of patients resolved BCVA reduction [32].

The exposure-safety analyses of DREAMM-2 evaluated the likelihood of G2 and G3 corneal AEs and their relationship with belamaf concentration. Ferron-Brady et al. [40] demonstrated that higher belamaf Ctrough (the predicted concentration on day 21 at the end of the first cycle) was associated with a higher probability of G2 or G3 corneal events or with an earlier onset of them. A history of dry eyes and lower baseline serum concentration of soluble sBCMA were associated with an increased risk of G2 and G3 corneal AEs. A history of dry eyes and baseline keratopathy prior to belamaf use was associated with a higher risk of any grade of blurred vision. Baseline keratopathy was also associated with an earlier onset of blurred vision along with an increased probability of grade ≥2 blurred vision.

The sub-study analysis of DREAMM-2 for the management of ocular toxicities demonstrated that prophylaxis with steroids eye drops as a mitigation strategy was ineffective as ocular prophylaxis [42]. It is strongly recommended the baseline ophthalmic examination prior to the first dose and then before each subsequent dose, even in the absence of symptoms. The use of a cooling eye mask, or vasoconstrictors prior to Belamaf infusion, is still unclear and should be used at the discretion of the physician. The use of preservative-free artificial tears is strongly recommended for all patients, at least four times a day from the day of belamaf infusion and throughout the treatment course [43].

Lin et al. [42] analyzed imipramine as a new potential drug against macropinocytosis in cellular and biological systems, so theoretically, the inhibition of belamaf macropinocytosis might reduce the occurrence of ocular toxicity, but the practical role of such inhibition is limited [44]. To date, it has been demonstrated that dose delay or dose reduction are the only actions to reduce ocular toxicity, in order to allow appropriate time for replacement of corneal epithelial cells. However, no specific guidelines are yet available to apply effective dose delay or reduction. Several trials are ongoing to evaluate alternative dose-reduction strategies. DREAMM-2 reported that treatment delay (more than 63 days) did not negatively impact belantamab mafodotin efficacy. Of 16 patients, 38% deepened their response; 38% maintained the same response, and 13% showed an increase of serum monoclonal protein in the absence of criteria for disease progression [45].

Considering the high frequency of ocular symptoms with Belamaf treatment, patients should be monitored closely with ophthalmology and hematology to ensure the effective and safe use of Belamaf.

Regarding hematologic toxicities, thrombocytopenia was reported as the most common adverse event during Belamaf treatment. In the DREAMM-1 study, grade 3–4 thrombocytopenia and anemia were reported in 35% and 15% of patients, respectively. This data was confirmed in the DREAMM-2 trial: grade 3–4 thrombocytopenia was observed in 22% of patients treated in the 2.5 mg/Kg arm compared to 32% in the 3.4 mg/Kg arm [32]. Furthermore, thrombocytopenia was reported in the DREAMM-6 study, leading to treatment discontinuation and dose delay in 33% and 39% of patients, respectively [45]. 

Other reported minor extra-hematological adverse events were pneumonia, GGT and ALT serum increase without liver dysfunction, hypertension, hypercalcemia and fatigue.

As previously mentioned, premedication before each belamaf infusion is not mandatory. In the DREAMM-2 study, premedication was not mandatory; specifically, it was administered in about a quarter of all patients, but it did not reduce the rate of infusion-related reactions (IRRs). IRRs were reported in 21% of patients treated in the arm of 2.5 mg/Kg, and 16% of patients in the arm of 3.4 mg/Kg experienced IRRs, mostly during the first infusion, and nearly all were grade 1–2.

## 3. Clinical Trials on Belantamab Mafodotin 

### 3.1. DREAMM-1, DREAMM-2 and DREAMM-3 Studies

The first human phase I trial evaluating belamaf monotherapy was the international Driving Excellence in Approaches to MM (DREAMM-1) study [41]. No maximal tolerated dose was found in the dose-escalation phase of the study in which 38 patients, 76% of whom had received at least five prior lines of therapy, received from 0.03 mg/kg to 4.6 mg/kg intravenous belamaf every 3 weeks until progression. The dose of 3.4 mg/kg was the RP2D chosen in the dose expansion phase in 35 patients, among whom 57% had received more than five prior lines of therapy, more than 90% were double-refractory and 40% were refractory to daratumumab. After a median follow-up of 12.5 months, patients enrolled in part 2 of the trial had an ORR of 60%, with 55% achieving at least VGPR. Belamaf monotherapy was found active in double- (ORR = 56.3%) and triple-refractory (ORR = 38.5%) patients as well in those with more than five prior therapies (ORR = 50%) [41]. The median time to first response was 1.2 months, the median duration of response was 14.3 months and the median PFS 12 months, being 6.2 months in triple-refractory patients. As regard safety profile, the most common grade 3–4 adverse events were thrombocytopenia (35%) and cornel events occurring in 69% of patients, most commonly blurred vision (51%), dry eye (37%) and photophobia (29%). After the DREAMM-1 study, pivotal phase II, open-label, two-arm, multicenter DREAMM-2 trial [32] led to the approval of belamaf in RRMM. This study enrolled RRMM patients who had received at least three prior lines of therapy, who were refractory to a PI and an IMiD and who were refractory or intolerant to an anti-CD38 mAb. Patients whose median age was 66 years were randomized to receive belamaf 2.5 (*n* = 97) or 3.4 mg/kg (*n* = 99) every 3 weeks until disease progression or unacceptable toxicity. As per baseline patient characteristics, patients received a median of 7 (3–21) and 6 (3–21) previous lines of therapy in the 2.5 mg/kg and 3.4 mg/kg cohort, respectively, with 100% of patients refractory to daratumumab in the 2.5/kg cohort in which approximately 27% of patients were at high risk cytogenetic, harboring t(4;14, t(14;16) or del(17p) and 23% had extramedullary disease. The primary endpoint DREAMM-2 trial was ORR and the final analysis [46] showed after a median follow-up of 12.48 months for patients enrolled in the 2.5 cohort and 13.77 months for those randomized to the 3.4 mg/kg cohort, an ORR of 32% and 35%, respectively. Among patients who obtained at least VGPR (19% in the 2.5 mg/kg group and 24% in the 3.4 mg/kg cohort), 36% and 23%, respectively, achieved MRD negativity. The median duration of response (DoR) and media PFS were 12.5 and 2.8 months in the 2.5 mg/kg cohort, and they were 6.2 and 3.9 months in the 3.4 mg/kg group. Notably, median PFS was higher in patients achieving high-quality response (≥VGPR), being 14 months and 16.8 months for 2.5 mg/kg and 3.4 mg/kg cohorts, respectively. As regard OS, the median estimate was 15.3 months in patients allocated to the 2.5 mg/kg arm, but it resulted in 30.7 months in patients achieving at least VGPR. In patients receiving 3.4 mg/kg estimated median OS 14 months, being 35.5 in those with ≥VGPR [46]. Among patients treated with belamaf 2.5 mg/kg, efficacy was documented either in patients with standard-risk cytogenetics, defined as patients with none of t(4;14, t(14;16), del(17p) or 1q21+, in whom ORR was 34% and median DoR was not reached after a median follow-up of 13 months or in patients with high risk cytogenetic (with any of the above abnormalities) in whom ORR was 29% and median estimated DoR 10.3 months. Moreover, patients with mild or moderate renal impairment obtained similar ORR compared with those with normal renal function [40].

In the DREAMM-2 study, the median time on therapy was 2.1 months (0.5–41 months) and 2.8 months (0.5–42.8) for patients randomized to 2.5 mg/kg and 3.4 mg/kg, respectively. As per the safety profile, grade ≥ 3 adverse events occurred in 84% of patients receiving belamaf 2.5 mg/kg and 83% of those treated with 3.4 mg/kg, requiring dose reduction in 36% and 44% and permanent discontinuation of belamaf in 9% and 5% of patients, respectively [46]. The main grade ≥ 3 hematologic toxicities were anemia occurring in 21% and 28% of 2.5 mg/kg and 3.4 mg/kg cohorts, whereas thrombocytopenia was documented in 19% and 29% of patients, respectively. Among nonhematologic adverse events, the most common was ocular toxicity, whose rate observed in patients allocated to 2.5 mg/kg was similar to that of the 3.4 mg/kg group. In the first cohort, any grade keratopathy occurred in 71% of patients (grade ≥ 3 = 29%), blurred vision in 25% and BCVA reduced to 20/50 or worse in 48% of patients. The median time to keratopathy resolution was 120 days, whereas the median time to resolution of the first BCVA event was 23 days. However, only 3% of patients in both study arms permanently discontinued treatment because of ocular events [46]. Based on results and the benefit-risk profile observed in the pivotal DREAMM-2 study, belamaf received accelerated approval on August 2020, at the dosage of 2.5 mg/kg every three weeks, for RRMM patients who have received at least four prior therapies, including an anti-CD38 mAb, a PI and an IMiD [47]. The ongoing phase II randomized DREAMM-14 trial is recruiting RRMM patients with ≥3 prior lines of therapy in order to evaluate if modifying the dose of belamaf monotherapy, the schedule or both a reduction in the incidence of ocular events, the primary endpoint of the study, can be achieved.

The phase III, open-label, randomized DREAMM-3 trial, comparing single agent belamaf to pomalidomide/dexamethasone (Pd) in 325 who had received at least 2 prior lines of therapy including lenalidomide and a PI, failed its primary endpoint since, despite a longer PFS for patients enrolled in the belamaf arm compared with Pd arm (11.2 months vs. 7 months), HR was 1.03 (95% CI: 0.72–1.47). However, at least VGPR was documented in 25% and 8% of patients receiving belamaf and Pd, respectively, and a safety profile was consistent with the one already known [48]. 

### 3.2. Ongoing Studies with Belantamab Mafodotin-Based Regimens in Relapsed/Refractory MM

Phase I/II Algonquin study evaluated the safety and efficacy of different doses and schedules of belamaf combined with Pd (Bela-Pd) in RRMM who had received at least one prior line of therapy, exposed to lenalidomide and PI and pomalidomide naïve [49]. In Part 1 of the study, a dose-escalation phase, patients were treated with pomalidomide 4 mg days 1–21, dexamethasone 40 mg weekly and belamaf as a single dose of 1.92 mg/kg every 4 weeks, as a single dose of 2.5 mg/kg every 4 weeks, every 8 weeks or every 12 weeks, or split on days 1 and 8 (2.5 mg/kg or 3.4 mg/kg) every 4 weeks. Considering the 54 triple-class exposed patients enrolled in the study, the median age was 67.5 years, the median number of prior lines of therapy was 3 (range 2–5) and 72.2% were triple-refractory. Across all cohorts, ORR was 86%, and 60% of patients achieved at least a VGPR. As per outcome measures, after a median follow-up of 5.7 months, the median PFS resulted in being 15.6 months, an impressive result considering that in the prospective LocoMMotion study, triple-class exposed patients treated with standard of care had a median PFS of 4.6 months [22]. Most common grade ≥ 3 adverse events included keratopathy (55%), neutropenia (37%), thrombocytopenia (27.5%) and decreased BCVA (23.5%). Other studies with belamaf also including patients with early relapse are ongoing. Phase I/II DREAMM-6 study is exploring the safety and activity of up to 3 dose levels and up to dosing schedules of belamaf in combination with lenalidomide/dexamethasone (Rd, arm A) or bortezomib/dexamethasone (Vd, arm B) in patients with ≥1 prior line of therapy. Preliminary results of 18 patients with a median of 3 prior lines of therapy (range 1–11) enrolled in arm B and receiving belamaf 2.5 mg/kg single dosing plus Vd showed an ORR of 78% with 67% of patients achieving at least VGPR. After a median of 25.5 weeks on treatment, the median DoR was not reached. Grade ≥ 3 thrombocytopenia occurred in 66% of patients, grade 3 keratopathy in 61%, whereas peripheral neuropathy (all grade ≤ 2) was observed in 33% of patients [45]. The international phase III DREAMM-7 trial is enrolling RRMM patients who have received at least 1 prior line of therapy, randomizing them between belamaf/bortezomib/dexamethasone (B-Vd) and daratumumab/bortezomib/dexamethasone (D-Vd) in. In the same set of patients, phase III DREAMM-8, aiming to enroll 450 patients with RRMM worldwide, is comparing belamaf/pomalidomde/dexamethasone (B-Pd) vs. pomalidomide/bortezomib/dexamethasone (PVd). In Table 1, we summarized other ongoing clinical trials, including belamaf in patients with RRMM.

### 3.3. Ongoing Studies with Belantamab Mafodotin-Based Regimens in Other Cancers

EMN27 phase 2 trial is recruiting patients with relapsed refractory AL Amyloidosis to receive Belantamab mafodotin at the same dose approved in MM, with the primary endpoint of CR/VGPR (NCT04617925). Another phase 1/2a study, not yet recruiting, has been built to enroll relapsed patients with AL amyloidosis (NCT05145816). A phase 1 trial is evaluating Belantamab mafodotin for the treatment of high-risk Smoldering Myeloma (NCT05055063). Belantamab mafodotin is on the study also in relapsed plasmablastic lymphoma and ALK+ large B cell lymphoma in a phase 2 trial (NCT04676360). Data are awaited.

### 3.4. Sequencing of Belantamab Mafodotin with Other BCMA-Targeting Immunotherapies

The availability of anti-BCMA immunotherapies as CAR-T cells products ide-cel [29] and cilta-cel [50], as well as the approval by FDA and EMA of bispecific antibody teclistamab [31], raised the question of the effectiveness of retreatment with another product in patients already exposed to BCMA targeting immunotherapies. A recent analysis, using data from DREAMM-1 and DREAMM-2 trials, evaluated free sBCMA concentration at baseline, at the achievement of best response and, at the latest, progression in order to study BCMA expression during treatment with belamaf [51]. In 97% of patients who responded to treatment and later progressed in the DREAMM-2 trial, the sBCMA level significantly decreased during response, returning to the baseline level at the time of progression, suggesting that target loss does not represent the main mechanism of resistance in these patients. In the recent study by Cohen et al. [52], 13 patients with advanced MM (previous lines of therapy = 8) received cilta-cel after belamaf, resulting in an ORR of 61.5%, with 71% of evaluable patients achieving MRD negativity. After a median follow-up of 11.8 months, the median DoR was 11.5 months, and the median PFS was 9.5 months. The opposite sequence can also be effective, as shown by Gazeau et al. [53], reporting a response that can be durable in patients receiving belamaf after CAR T cell therapy.

### 3.5. Belantamab Mafodotin in Newly Diagnosed MM

As well as for other novel immunotherapies, several clinical trials are exploring belamaf in the upfront setting. At the last ASH Meeting, preliminary results from the phase II Spanish GEM-BELA-VRd trial were presented [54]. Treatment included six induction cycles with VRd plus belamaf 2.5 mg/kg every 8 weeks, high dose melphalan and ASCT, followed by consolidation with two cycles with Bela-VRd and maintenance with lenalidomide until progression plus belamaf for 2 years. Among the 40 patients who received at least four cycles of induction, ORR was 82%, at least VGPR 69% and CR 13%. Ocular toxicity represented the most common toxicity since 77.5% of patients developed blurred vision and 60% any grade keratopahy. Neutropenia and thrombocytopenia were the most frequent hematologic adverse events. In the same Meeting, the Greek Group presented preliminary results of 36 transplant-ineligible patients (median age = 72.5 years) treated with the regimen Bela-Rd in the phase I/II study [55]. ORR was 97%, 72% of patients achieved at least VGPR and all patients were in response after a median follow-up of 9.5 months. Grade 3 visual acuity reduction occurred in 33% of patients, whereas no patients developed grade 3–4 keratophaty as well as grade 3–4 thrombocytopenia was not documented. In Table 2, we summarized the main ongoing clinical trials with belamaf in NDMM.

## 4. Real Life Data on Belantamab Mafodotin

### 4.1. USA Real-Life Experiences

Several American real-life experiences have been published in order to better define the efficacy and toxicity of belantamab outside clinical trials, which had restricted patients’ selection, risking providing poorly reproducible data in real-life settings (Table 3). 

Vaxman et al. [56] retrospectively identified 36 MMRR patients who received at least one dose of belantamab outside a clinical trial at all three Mayo Clinic sites, from September 2020 to June 2021, with a median follow-up of 6 months. Patients’ median prior lines of therapy were eight, more than DREAMM-2 trial, whose six patients (17%) received belantamab in combination with other agents (pomalidomide, cyclophosphamide and thalidomide) and seven patients (19%) had already received another anti BCMA agent (CART) prior to belantamab therapy. These features characterized a heavily pretreated population, also BCMA-refractory, differently from the pivotal study. The median time from diagnosis to the first belantamab dose was 7 years. They reported an ORR of 33% with a median PFS and OS for the whole cohort of 2 months and 6.5 months, respectively. Eighteen patients died, all for disease progression. As for safety, keratopathy developed in 16 patients (44%); it was grade 1 in six patients, grade 2 in seven patients and grade 3 in three patients. Six patients reported decreased visual acuity. Two patients had grade 3 infusion-related reactions, and two had infections. The hospitalization rate was 33%, mostly related to MM complications (malignant ascites, hypercalcemia, pain management). The only previously unreported toxicity was seen in one patient hospitalized due to a suspected TLS. Five patients are still on therapy; treatment discontinuations were in 85% of patients because of disease progression in 28 patients (77%) and keratopathy in three patients (8%). Despite the low number of enrolled patients, this was one of the first papers which confirmed the efficacy and safety data of clinical trials in real-world settings [56].

More recently, Abeykoon et al. [57] conducted a retrospective observational study on 38 RRMM patients treated with belantamab at Mayo Clinic between January 2020 and January 2021, with a median follow-up of 11 months, to evaluate the impact of belantamab mafodotin-induced ocular toxicity on the outcomes of patients themselves. Authors found a 29% ORR, with a median PFS and OS of 2 and 7.2 months, respectively, in a really heavily pretreated population (median prior lines of therapy 8, range 2–15), with 89% of patients having high-risk disease characteristics and a median time from diagnosis to belamaf start of 7 years (range 1–19.7). Ocular toxicity was observed in 27 (75%) patients: keratopathy in 25 (69%), decreased BCVA in 21, and/or ocular symptoms like xerophthalmia in 13 (36%). Ocular toxicity seemed to develop earlier in responders patients than in non-responders. Authors found that keratopathy significantly complicated belamaf therapy mitigating its full potential effectiveness. For ocular toxicity, belantamab was permanently discontinued in five (14%) patients after a median of three doses (range: 2–3), delayed in nine (25%) patients and reduced in doses in four (11%) patients. Belantamab discontinuation rate seemed to be higher than in the DREAMM-2 trial, probably because the longer follow-up may have captured more patients with keratopathy-associated discontinuation. Moreover, only a few patients maintained their response (PR, 2 and VGPR, 2) during the prolonged keratopathy-related treatment interruption, compared to the 88% of patients maintaining their response in DREAMM-2, probably because real-world population may be more heterogeneous, more heavily pretreated with much more high-risk diseases [57]. 

Becnel et al. [58] presented at ASCO 2022 data of a retrospective, single-center, real-world experience of belantamab mafodotin in 39 RRMM patients treated between November 2020 and November 2021 at MD Anderson Cancer Center in Houston. The overall population had seven median prior lines of therapy (range 3–16) and 38% of whom had high-risk FISH features, 38% extramedullary disease and 69% did not meet eligibility criteria for the DREAMM-2 study, picturing a more difficult population to treat than the pivotal study. Moreover, eight patients were even anti-BCMA-refractory; one of them obtained a PR and another an MR. The authors found a 27% ORR with a CBR of 35%. Median PFS was 1.8 months and median OS 9.2 months, with a not reached median DOR at a median follow-up of 10.1 months. Twenty-five (76%) patients reported keratopathy; BCVA was described in 75% of patients, with a median time to a first ocular adverse event of 1.3 months. However, there was no information about the recovery of these events. The authors specified that the most common reason for treatment discontinuation was disease progression in 75% and adverse events in 9% of patients. This retrospective study demonstrated a good efficacy and safety profile of belantamab in a real-life population not eligible for clinical trials [58]. 

Hultcrantz et al. [59] presented data at ASH Meeting 2022 from a retrospective, single-center, observational study of 90 RRMM patients who received at least one dose of commercial belantamab mafodotin at Memorial Sloan Kettering Cancer Center between October 2020 and October 2022. Their median prior lines of therapy were six (range 2–14), and 19% of them were already BCMA-exposed (12 CART, 6 bispecific antibodies and 2 belantamab). No data were available about the frailty of enrolled patients. ORR was 42%, with 36% having disease progression as the best response, similar between BCMA-exposed and BCMA-naïve. Median PFS was 4 months with a median DOR of 13.1 months and median OS of 20.5 months. Ocular toxicity was reported in 58 (64%) patients globally, being keratopathy in 57 (63%) patients and BCVA reduction in 41 (46%) patients. Twenty-six (29%) patients reduced belantamab doses, 24 (27%) delayed doses and 9 (10%) discontinued treatment for ocular toxicity. No new safety concerns were identified in this real-life population; the majority of patients were able to continue belantamab and maintain a clinical response [59]. Hultcrantz et al.also presented at the same meeting retrospective data of 137 RRMM patients who had ≥1 record for belantamab administration, from the US EHR-derived Flatiron Health Database, from January 2011 to December 2021. Among them, there were 40 (29.2%) penta-refractory patients; 64.2% of all patients had received at least five lines of therapy prior to belamaf start, being globally less pretreated than patients in DREAMM-2. The median time from MM diagnosis to the first belantamab administration was 4.8 years, shorter than in the DREAMM-2 study (5.49 years). But patients had a lot of comorbidities; more than 50% of patients had cardiovascular disease, 24% had cardiac disease and 38% had renal disease, but performance status was not reported. ORR was 30.2% at six months, similar to the pivotal DREAMM-2 study, and median PFS was 5.4 months, slightly longer. Among the whole population, 51.8% of patients had ocular toxicity (40.9% keratopathy, 64.2% blurred vision, while a reduction in BCVA rate was not specified). Most ocular adverse events (58.5%), including keratopathy events (76.0%), were managed by therapy hold. The most common reasons for treatment discontinuation were disease progression (33.8%), treatment toxicity (19.7%) and a combination of both (15.5%). Authors confirmed in a numerous real-world cohorts of RRMM patients, slightly older than in DREAMM-2 (median age: 68 vs. 65 years) and with multiple comorbidities, efficacy and safety data from the pivotal study, suggesting that belantamab could help fill the unmet treatment need in this setting [60]. 

### 4.2. Asian Real-Life Experience

Shragai et al. published data from an Israelian observational multicenter real-life study treating 106 RRMM patients with belantamab between 2019 and 2021, during the compassionate program, with or without steroids. Baseline population characteristics were similar to the DREAMM-2 study, but patients had lower median prior lines of therapy (6 vs. 7). Differently from the pivotal study, there was 32% of penta-refractory patients, and the median age was a little higher (69.4 vs. 65 years). ORR was 45.5%, and this rate was maintained among different subgroups (age, sex, triple-or penta-refractoriness, ISS, R-ISS, high-risk cytogenetics, EMD). At a median follow-up of 11.9 months, the median PFS was 4.7 months in the whole population, while it was 8.8 months in responders, confirming the significant association between response deepness and outcomes already reported in DREAMM-2. There was no difference in PFS between the whole cohort and triple-refractory or penta-refractory patients. Median DOR was 8.1 months and median OS 14.5 months, without differences based on cytogenetic risk or refractoriness status, but better in responders than not responders. As for safety, the most common adverse event was ocular; 65 (68.4%) patients experimented with keratopathy, of whom 63.4% resolved to grade 1 or less. Ocular symptoms were reported in 36.8% of evaluable patients. Discontinuations due to ocular toxicity were four (3.8%), similar to DREAMM-2, and delays were 82 (70.7%), without correlation between the proportion of dose delay and response rate. Other safety concerns were thrombocytopenia in 29 (27.4%) patients, 17.9% grade ≥ 3, infections (11.3%, with 2 cases of hepatitis B reactivation) and anemia (11.3%). Considering that these patients had fewer prior lines of therapy, more patients with >75 years (23% vs. 13%) and the presence of penta-refractory patients (32 vs. 0) than DREAMM-2, without patients with severe renal failure or cytopenic because they were excluded from the compassionate program, ORR seemed to be higher in the real world experience, with similar outcomes. Interestingly, both ORR and outcomes did not seem worse in penta-refractory patients than the overall cohort, encouraging anti-BCMA retreatment in the challenging setting of penta-refractory patients. No new safety signals were reported in real life, and the authors confirmed the reversibility of ocular findings. This paper reported for the first time two cases of tumor lysis syndrome, highlighting the need to select patients who could receive adequate prophylaxis [61]. 

### 4.3. European Real-Life Experiences

Offidani et al. reported data from an observational, multicenter, retrospective real-life study on 67 RRMM patients treated with belantamab in compassionate use programs such as Named Patient Program (NPP) and Expanded Access Program (EAP) in different Italian centers under the aegis of European Myeloma Network (EMN). Compared to DREAMM-2, this cohort had fewer previous lines of therapy (5 vs. 7) but similar median age with similar general characteristics. Authors found similar ORR (31%) and CBR (37%). Median PFS was 3.7 months, median OS 12.9 months and median DOR 13.8 months; they seemed higher than the pivotal study. Authors confirmed that ocular toxicity was the most common, mostly grade < 3 (87%), with keratopathy reported in 23 (74%) patients, ocular symptoms in 5 (16%) and changes in BCVA in 3 (10%), all reversible during the follow-up. Moreover, the drug was discontinued in 45% of cases, and in 13% of cases, it needed only a dose reduction. Thrombocytopenia was the second most common adverse event, reported in 14 (87.5%) patients, always reversible. Infections, infusion reactions and one case of secondary gallbladder cancer were described as less frequent adverse events. The most common cause for drug discontinuation was disease progression (75%). Authors demonstrated in explorative univariate analysis that PFS was negatively affected by patient characteristics, as age > 65 years (*p* = 0.094) and ECOG ≥ 2 (*p* = 0.012), rather than classical prognostic features, paving the way for future investigations on the best applicability of this drug [62]. 

Iula et al. [63] published data of an observational, multicenter, real-life study on 28 RRMM patients treated with belantamab in four Hematology Units of the Campania region in Italy. Their median prior lines of therapy were six, and their general characteristics were quite similar to DREAMM-2 patients, except for the number of patients with severe renal failure (9% in the evaluable population vs. 2% in DREAMM-2). Authors reported 40% ORR without difference in different subgroups based on renal failure severity. Interestingly, patients with renal impairment were 20 (71%). They were classified based on the severity of renal failure in eight patients with mild dysfunction (60 ≤ GFR< 90 mL/min), 8 with moderate (30 ≤ GFR > 60 mL/min) and 4 with severe (GFR < 30 mL/min), that were slightly higher rates than DREAMM-2. Patients with mild, moderate or severe renal failure showed an ORR of 50%, 25% or 50%, respectively, similar to that reported in DREAMM-2 and Mayo Clinic trials, even considering the special population of the DREAMM-2 trial with mild (49%), moderate (25%) and severe (2%) renal dysfunction. These results are really important because patients with severe renal dysfunction are always poorly represented in published clinical trials, suggesting belantamab could be safely administered in MM patients with severe renal impairment without reducing clinical benefits, probably because the drug is mainly degraded and eliminated through internalization and intracellular proteolysis [64]. Median PFS and OS were 3 and 8 months, respectively, at a median follow-up of 6.5 months. Median DOR was not reached. Selecting the overall population by the best response, median PFS was not reached in the subgroup who obtained a VGPR (vs. 11 months in patients with PR), endorsing the association between deep responses and better outcomes, even in real-life settings. ORR remained the only factor impacting outcomes in univariate analysis. Thrombocytopenia was the most frequent adverse event (46%), mostly grade < 3 (14% grade 3–4), followed by keratopathy (32%) that was always reversible and grade ≥ 3 in only 11% of cases, leading to drug discontinuation. The authors describe clinical ocular management of patients, which provided an ophthalmological visit only in case of ocular problems during the course of treatment, except for one center where the ophthalmological visit was made every three months after starting belantamab administration. The authors specified that there were no standardized procedures across different centers for keratopathy diagnosis and management, and this could have underestimated the real rate of silent corneal damage. This is very important in a real-life study because it better mirrors real-life clinical management, as a close interdisciplinary assessment could be difficult to organize based on ophthalmologist availability and patients’ clinical conditions [63].

De La Rubia et al. presented at the ASH Meeting 2022 data from an observational, retrospective, and multicenter study which included 126 RRMM patients from 59 centers who received at least one dose of belantamab within compassionate use or expanded access programs in Spain between November 2019 and June 2021. The overall population was older than DREAMM-2 one (median age 72.5 vs. 65 years), with severe renal failure in more cases (8% vs. 2%), more extramedullary disease rate (31.4% vs. 23%) and 34.6% of patients were penta-refractory. However, median prior lines of therapy were less than the clinical trial (5 with a range from 4 to 6 vs. 7). ORR was 46.4%, really higher than in DREAMM-2, and it was maintained in triple-refractory and penta-refractory patients. Likewise, there were no differences in response rates among different age subgroups. Median PFS and OS were 3.6 and 11.1 months, respectively, at a median follow-up of 13 months in the overall population. The authors also demonstrated a significant correlation between outcomes and deepness of response, like in DREAMM-2, being median PFS 14.4 vs. 1.6 months and OS 23.3 vs. 3.9 months, in patients who obtained a response ≥MR vs. <MR, respectively. In patients achieving at least MR, the median DoR was 13.9 months. Ocular adverse events were reported in 53.2% of patients, with keratopathy being the most frequent (46.8%), in most cases grade < 3 (82%), causing drug discontinuation in only two patients. Thrombocytopenia was described in 15.4% of patients, being of grade ≥3 in 10.9% of cases. Infections were the most important non-hematological toxicity; it was reported in 15% of cases, 5.6% of whom grade ≥ 3. In conclusion, Spanish real-life experience documented an efficacy and safety profile similar to the pivotal trial, with slightly higher ORR, even if it was used in a cohort of patients with really fewer median lines of prior therapies but with more aggressive disease characteristics [65]. Roussel et al. presented at the ASH Meeting 2022 results of the ALFA study, a non-interventional, retrospective study of 184 RRMM patients who started belamaf in 46 centers in France during early access programs from April 2020 to June 2021. The study population was older than DREAMM-2 one (median age 70.3 vs. 65 years) with more frailty features (12.3% vs. 2% of patients with severe renal failure were included, 11.5% of patients with ECOG ≥ 3). Importantly, 78.8% of patients were penta-exposed, without information about refractoriness status. Fifty-eight percent of patients received ≥5 prior lines of therapy, lower than in DREAMM-2. ORR was 32.7%, with a median CBR of 36.4%. Interestingly, both ORR and CBR were 0% in the subgroups of patients with the extramedullary disease (only 15 patients). Median PFS was 2.4 months in the overall population; when it was stratified by the best response, it was 20.6 months in patients with ≥VGPR, 7.1 months in patients with PR, and 1.6 months in others, demonstrating the significant correlation between outcomes and deepness of disease response again. The same was for median OS, which was 8.8 months for the full cohort of patients, 17.5 months in patients with PR, 14.1 months in MR, 9.5 months in SD and 5.6 months in the other patients; median OS in progressive disease was drastically lower (3.3 months). The most frequent adverse events were ocular (56%), most of them grade < 3 (71.5%). Keratopathy was reported in 41.8% of patients, only 8.2% grade ≥ 3, decreased visual acuity in 10.9% and other ocular disorders in 13%. Ocular adverse events caused permanent drug discontinuation in 12.5% of cases, dose modification in 19.6% and temporary interruption in 11.4%, with a median duration of delay of 32 days. Thrombocytopenia occurred in 13.6% of patients and infusion reactions were reported in 3.3% of patients. The results of the ALFA study were similar to those of the DREAMM-2 trial, even if the population was older and more frail but less pre-treated [66]. 

More recently, Talbot et al. [67] published results from IFM 2020-04 real-world study on the efficacy and safety of Belantamab in 106 RRMM, based on data from the nominative ATU (authorization temporaire d’utilisation) in France, from November 2019 to December 2020. Their median age was 66 years, and their median number of prior lines of therapy was five (range 3–12). All patients were triple-exposed, but only 55.6% were triple-refractory, and there were 11.3% of penta-refractory patients, which were the two most relevant differences between this and the DREAMM-2 study. So, ORR was 38.1%, higher than the DREAMM-2 trial, but outcomes remain similar with a median PFS of 3.5 months, OS of 9.2 months and DOR of 9 months. Interestingly, in subgroups analysis, no significant difference was observed in terms of outcomes based on the presence of extramedullary disease, cytogenetic risk or refractoriness status, except a significant difference in OS between fit and unfit patients (OS 16.8 vs. 5 months, *p* = 0.01), probably not only related to belantamab effect. This paper confirmed the significant correlation between outcomes and deepness of response, with an HR of 3.05 for OS and 2.91 for PFS. Ocular toxicity was confirmed to be the most frequent (48%), grade 3 in 40.8% of patients, and keratopathy was reported in 37.5% of patients. Ocular events resulted in delayed treatment administration or in a dose reduction in 30% of patients. Thrombocytopenia was the second most relevant toxicity, reported in 43.8% of patients grade ≥3. No new concerns were highlighted in this real-life experience.

### 4.4. A Comparison of Different Real-Life Experiences

Despite there being no significant differences in efficacy or safety among different geographical settings, a trend of better outcomes could be identified in European and Asian experiences. These studies have sometimes reached a 35–40% of ORR even without an increase of PFS and DOR, probably because they are more recent and they enrolled fewer pretreated patients. As for safety, the discontinuation rate for toxicity, which could be considered the best representative parameter, remains often similar among different papers, describing a good safety profile of the drug even in real-life populations.cancers-15-02948-t003_Table 3Table 3Real life Belantamab Mafodotin experiences.TitlePatients (n)Population: Median of Prior Lines of Therapy (Range), Median Age (Range)Outcomes (ORR, mPFS, mDOR)Safety: Keratopathy Grade ≥ 3 (%)Therapy Discontinuation for Toxicity (%)***USA experiences***[56]368 (7–11)61 (37–83)33214.388[57]388 (2–15)67 (49–90)29231414[58]397 (3–16)66 (39–89)271.8NR129[59]906 (2–14)68 (37–88)42413.11610[60]1375 (4–7)68 (±10)30.25.4-38.619.7***Asian experiences***[61]1066 (2–11)69 (36–88)45.54.78.140-***European experiences***[62]67566 (42–82)313.713.81345[63]286 (3–14)67.5 (51–83)403NR,1111[65]1565 (4–6)72.5 (64–77)46.43.613.917.97.9[66]184570 (63–76)32.72.4-8.212.5[67]1065 (3–12)66 (37–82)38.13.2937.5 (overall)-

## 5. Conclusions and Future Directions

The increasing number of patients receiving all classes of drugs as IMiDs, PIs and mAbs, acquires multi-refractoriness status and requires the development of novel therapeutic strategies that try to overcome the clonal complexity and heterogeneity of MM. Additionally, naked mAbs, among new immunotherapies, Belantamab Mafodotin has been the first-in-class anti-BCMA ADC to be approved for advanced RRMM. In the DREAMM-2 study [32], belamaf monotherapy was able to induce substantial ORR in patients who had received more than five prior lines of therapy, showing activity either in standard or in high-risk RRMM patients [46], with a remission duration of quite one year. Notably, these results have also been confirmed in real-world experiences. Impressive anti-myeloma activity has been documented with bispecific antibodies and CAR T cell therapies, but these immunotherapies are not yet available in many countries and, concerning CAR T cells, they required personalized manufacturing time with a median period from leukapheresis to infusion for ide-cel of 40 days [29], making this therapy challenging in patients with rapidly progressive disease. Other limitations should be taken into account as the presence of an adequate composition of T cell populations in the patients, the need for bridging therapy and good performance status, the need for family support and, last but not least, the very high cost of therapy. Moreover, both CART cells and bispecific antibodies such as teclistamab, the only one approved for RRMM, necessitate hospitalization to manage possible early severe toxicities such as CRS (cytokine release syndrome) and ICANS (immune effector cell associated neurotoxicity syndrome). The main advantages of using belamaf in triple-refractory patients are the immediate availability and no need for hospitalization. Moreover, while retreatment with anti-CD38 mAb is not effective, the other anti-BCMA immunotherapies can be administered after belamaf leading to a significant duration of response in very heavily pretreated MM patients [52]. Ocular toxicity represents a peculiar side effect of belamaf, but close monitoring and collaboration between hematologists and ophthalmologists are making this toxicity easier to predict and manage. Several ongoing studies are evaluating different belamaf doses and schedules in order to reduce its incidence. In RRMM, the results of trials exploring belamaf in combination with IMiDs and PIs, as well as those including very innovative agents, are expected since these potential new regimens could represent an exciting step for belamaf to address unmet needs in MM paradigm. Moreover, very encouraging results have been reported with triplet Bela-Pd in early relapse so within the next years we expect that this regimen and others under evaluation enter the treatment landscape of less advanced RRMM. Belamaf is under evaluation also in AL amyloidosis and we hope that it can become a resource for patients affected by this very difficult to treat disease. It is desirable that Exploring lower doses of belamaf translates into an improved tolerability in order to prolong therapy and consequently to obtain a long-lasting response.

## Figures and Tables

**Figure 1 cancers-15-02948-f001:**
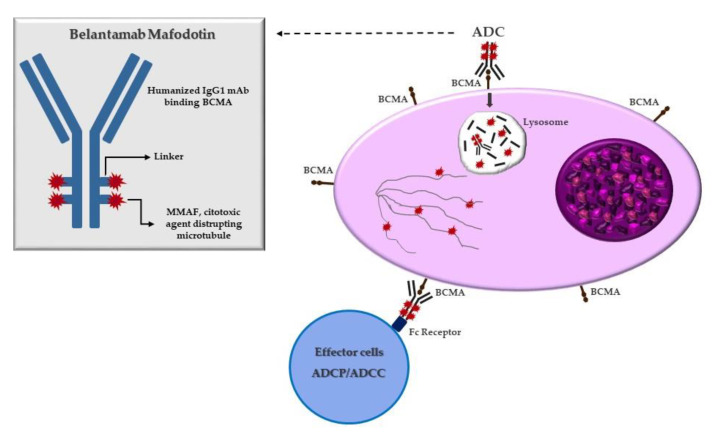
Mechanism of action of belantamab mafodotin. After binding to BCMA on plasmacell, ADC is degraded in the lysosome with release of MMAF that leads to G2/M arrest and caspase 3 dependent apoptosis. Belamaf exerts antibody-dependent cell-mediated cytotoxicity (ADCC) and antibody-dependent cellular phagocytosis (ADCP).

**Table 1 cancers-15-02948-t001:** Other ongoing clinical trials, including belamaf in patients with RRMM.

Trial	Phase	Population	Intervention	Trial ID
Belantamab mafodotin + daratumumab, pomalidomide and dexamethasone in patients with one prior line of therapy and lenalidomide refractory	I/II	RRMM with previous 1 line of therapy and lenalidomide-refractory	Bela-DPd	NCT05581875
Belantamab mafodotin in combination with Kd for patients refractory to lenalidomide	I/II	RRMM with 1–3 prior lines of therapy, refractory to lenalidomide	Bela-Kd	NCT05060627
Belantamab mafodotin maintenance therapy after salvage autologous hematopoietic cell transplantation	I	RRMM with ≥2 prior lines, including PI, IMiD and anti-CD38 mAb	Belamaf	NCT05065047
Belantamab mafodotin with carfilzomib, pomalidomide, and dexamethasone	II	RRMM with ≥2 prior lines, including PI, IMiD and anti-CD38 mAb	Bela-KPd	NCT05789303
DREAMM-12	I	Renal impairment RRMM with ≥2 prior lines, including PIs and IMiDs	Belamaf	NCT04398745
DREAMM-13	I	Hepatic impairment RRMM with ≥2 prior lines, including PIs and IMiDs	Belamaf	NCT04398680
Belantamab mafodotin and nirogacestat in people with MM that has not responded to treatment or have come back after treatment	I	RRMM with ≥3 lines, including PI, IMiD and anti-CD38 mAb	Belamaf + Nirogacestat	NCT05556798
DREAMM-4	I/II	RRMM with ≥3 lines, including PI, IMiD and anti-CD38 mAb	Belamaf + Pembrolizumab	NCT03848845
DREAMM-5	I/II	RRMM with ≥3 lines, including PI, IMiD and anti-CD38 mAb	Belamaf + innovative drugs	NCT04126200
DREAMM-20	I/II	RRMM with ≥3 lines, including PI, IMiD and anti-CD38 mAb	Bela-xRd. X will be either a SoC or an emerging treatment	NCT05714839
Belantamab mafodotin, cyclophosphamide, and dexamethasone	I/II	RRMM with ≥3 lines, including PI, IMiD and anti-CD38 mAb	Bela-Cd	NCT04896658
Belantamab mafodotin and elotuzumab to enhance therapeutic efficacy in multiple myeloma	I/II	RRMM with ≥3 lines, including PI and IMiD	Belamaf + elotuzumab	NCT05002816
Master Protocol	I/II	RRMM with ≥3 lines, including PI, IMiD and anti-CD38 mAb	Belamaf + novel agents	NCT04643002
EMBRACE	II	RRMM with ≥3 lines, including PI, IMiD and anti-CD38 mAb	Maintenance with Belamaf after anti-BCMA CAR T cell therapy	NCT05117008

**Table 2 cancers-15-02948-t002:** Main ongoing clinical trials with belamaf in NDMM.

Trial	Phase	Population	Intervention	Trial ID
Belantamab mafodotin as pre- and post-autologous stem cell transplant and maintenance	II	Transplant eligible	Belamaf pre (day-42) ASCT and after for 2 years	NCT04680468
Belantamab mafodotin and lenalidomide in patients with minimal residual disease positive after stem cell transplant	II	Transplant eligible	Belamaf + lenalidomide for 6 cycles if MRD positivity after ASCT	NCT04876248
Belantamab mafodotin, pomalidomide and dexamethasone in high-risk MM	II	Transplant eligible High-Risk cytogenetics	Bela-Pd maintenance after ASCT	NCT05208307
Belantamab mafodotin in combination with daratumumab, lenalidomide and dexamethasone in transplant-ineligible patients	I/II	Transplant ineligible	Bela-DRd	NCT05280275
Belantamab mafodotin in combination with lenalidomide, dexamethasone and nirogacestat in transplant ineligible patients	I/II	Transplant ineligible	Bela-Rd + nirogacestat	NCT05573802
DEAMM-9	I	Transplant eligible and ineligible	Bela-VRd followed by Bela-Rd	NCT04091126
Belantamab mafodotin with carfilzomib, lenalidomide, dexamethasone	I/II	Transplant eligible and ineligible High-Risk cytogenetics	Bela-KRd	NCT04822337

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
