# Peer review of "Belantamab Mafodotin: From Clinical Trials Data to Real-Life Experiences"

_cancers, 2023, doi:10.3390/cancers15112948_

Round 1
Reviewer 1 Report
In the manuscript, the authors reviewed Belantamab Mafodotin, a novel ADC agent, used for Multiple Myeloma from clinical trial data to real life experiences. The overall structure of this review is logical and clear. The authors described Multiple myeloma, Belantamab Mafodotin, Clinical trials on Belantamab Mafodotin and Real life data on Belantamab Mafodotin. Such review is novel and helpful for the application of Belantamab Mafodotin in treating MM patients from safety, efficacy and real life insight. However, the manuscript needs to be revised before publishing in Cancers.
1. Some figures, such as the molecular structure of Belantamab Mafodotin, the schematic diagram of the Mechanism of action, etc., need to be added to the article, which could facilitate a more intuitive understanding of Belantamab Mafodotin.
2. In the introduction section, Belantamab Mafodotin was not introduced, I recommend that the Introduction be reorganized to briefly present the Belantamab Mafodotin first, and then describe others. For example, the introduction of “the addition of daratumumab” and “SLAMF7” is not very logical. Also, the introduction section is too long and needs to be well organized.
3. In lines 342 and 343, is the description micropinocytosis or macropinocytosis? And there is no causality between these two senescence.
4. The Future Directions section needs to discuss more potential indications from either Preclinical or Clinical insights.
5. Are there any differences of real life experiences between USA, Asia, and Europe overall? For example, which region has the best outcomes or safety?
6. The paragraph starting from line 236 should be moved forward.
7. In the section of Belantamab Mafodotin, the authors need to add a subsection to describe some recent laboratory research for other potential cancer (if possible).
8. Some latest references need to be reviewed. For example, Talbot A, Bobin A, Tabone L, Lambert J, Boccaccio C, Deal C, et. al. Real-world study of the efficacy and safety of belantamab mafodotin (GSK2857916) in relapsed or refractory multiple myeloma based on data from the nominative ATU in France: IFM 2020-04 study. Haematologica. 2023 Apr 20. doi: 10.3324/haematol.2022.281772. Epub ahead of print. PMID: 37078253.
Author Response
Response to Reviewer 1:
1. We have attached Figure 1 as requested to explain the mechanism of action of belantamab.
2. We have shorten the introduction and we have focalized several points as requested.
3. We have corrected lines 342 and 343, as requested..
4. We have discussed in the Future Directions section more potential indications from either Preclinical or Clinical insights as requested.
5. We have included a little chapter to highlight any differences of real life experiences between USA, Asia, and Europe, despite there are no great differences.
6. The paragraph starting from line 236 have been moved forward.
7. We have added a subsection to describe some ongoing laboratory research for other potential cancer (lymphoma, amyloidosis and smoldering myeloma).
8. We have reviewed references as requested.
Reviewer 2 Report
The manuscript entitled: “Belantamab Mafodotin: from clinical trials data to real life experiences” addresses a topic of interest in the area of ​​oncology, namely the treatment of multiple myeloma, reviewing published clinical trials and actual cases. I think it can give an important contribution to the area
Nevertheless, I would like to make a few remarks:
-The introduction could be shortened and complemented with figures
-Tables could be more informative. Thus, a reduction in the text could be possible, making the reading of the manuscript easier.
-Excessive use of abbreviations makes reading of the manuscript confuse
-The mechanisms of action of Belantamab Mafodotin could be resumed in a figure
English language needs same minor editing. In my opinion authors should use more concise language, with shorter sentences
Author Response
Response to Reviewer 2:
1. We have shortened the introduction and added a figure about the mechanism of action of belantamab (this point also answers to the last point).
2. We have corrected tables, shortening the text inside.
3. Abbreviations in the manuscript now seem not excessive to understand the meaning.
Reviewer 3 Report
I have reviewed the article " Belantamab Mafodotin: from clinical trials data to real-life experiences" The authors did well in reviewing the literature and providing the current clinical notes. They could include graphical abstracts to make the content easy for readers to follow.
Author Response
Response to Reviewer 3:
We have attached a graphical abstract as requested (below the abstract).
Round 2
Reviewer 1 Report
The authors carefully revised their manuscript, and it chould be accepted.
Reviewer 2 Report
The revised version of the manuscript seems to be suitable for publication